# BIM-Based AR Maintenance System (BARMS) as an Intelligent Instruction Platform for Complex Plumbing Facilities

**Pei-Huang Diao** and **Naai-Jung Shih** *

Department of Architecture, National Taiwan University of Science and Technology, 43, Section 4, Keelung Road, Taipei 106, Taiwan; diaoph@msn.cn
* Correspondence: shihnj@mail.ntust.edu.tw; Tel.: +886-02-2737-6718

**Abstract:** The traditional architectural design of facilities requires that maintenance workers refer between plumbing layout drawings and the actual facilities in complex, hidden, sealed in-wall, or low illumination environments. The purpose of a Building information modeling-based Augmented Reality Maintenance System (BARMS) in this study was to provide a smartphone-based platform and a new application scenario for cooling tower and pipe shutdown protocol in a real-world old campus building. An intelligent instruction framework was built considering subject, path, and actions. Challenges and solutions were created to monitor the subject and maintenance protocol while moving from inside to outside, between bright and dark environments, and when crossing building enclosures at roof level. Animated instruction using AR was interactive and followed the knowledge and management protocols of associated instruction aids. The results demonstrated a straightforward mapping of in-wall pipes and their connected valves, with practical auxiliary components of walking direction and path guidance. The suggested maintenance routes also ensured a worker's safety. Statistical analysis showed a positive user response.

**Keywords:** markerless augmented reality; intelligent instruction; smartphone app; architectural facility maintenance; building information modeling

## 1. Introduction

Complex organizations of building spaces usually create maintenance difficulties due to the design of mechanical, electrical, and plumbing (MEP) systems. This complexity not only necessitates expertise to fully comprehend design drawings, but also specialized knowledge to document and maintain a diversity of mechanical components and equipment. Since building spaces feature different levels and layouts with the incorporation of various parts, intensive design efforts are required to make optimal use of the limited space available for both the building systems and its occupants. The relationship between the original design information and its orientation in real spaces is frequently imperfect. Thus, it can be difficult to identify a perfect way to confirm the relative location of pipes that originate from the other side of a wall.

The integration of building component information and on-site maintenance is critical for as-built facility management (FM). In reality, accurately determining the locations of building components is characterized by difficulty and ambiguity. Since building design data are not updated frequently, using original design drawings to troubleshoot on-site situations becomes very challenging, especially when the specific maintenance work necessitates additional effort or experience. For monitoring or inspection after a building is occupied, a thorough comprehension of the corresponding operational manual is usually required, as is a well-controlled operational procedure in combination with hands-on

experience in a real site. Although 3D models and attributes defined in building information modeling (BIM) contribute to complete facility information, which can be accessed by maintenance workers from cloud computing services, in reality, the connectivity of piping layouts and the exact points of penetration are very difficult to follow from one space to another and beyond partitions. This difficulty can be increased when inspections have to be made across systems, where different levels of accuracy and symbolic representations exist.

Allocation of maintenance space, which constitutes an important safety concern, should be planned in advance and clearly indicated in building design diagrams afterward. Due to the complicated layout of mechanical equipment and the limited availability of nearby free space, space to access pipes may not be sufficiently large. In order to prevent unexpected injury, a route with clear signage and free of unanticipated obstacles should be provided with an appropriate visibility related to the updated location of the maintenance space.

A methodological approach to test BIM modelling needs to be applied from an empirical experiment of system design, test, and evaluation. Augmented reality (AR), using piping geometries facilitated by and integrated with both information and reality, can project BIM models into an environment for maintenance workers to follow. An AR system is needed to illustrate the application and integration potential of BIM in facility management, considering the co-relation problems between different systems and their operations in a real as-built environment. A cooling tower and piping maintenance case that applies simultaneous localization and mapping (SLAM) technology in positioning should also be investigated in order to support BIM-based AR maintenance troubleshooting. A smartphone-based system, which is convenient, should be investigated to provide functions that support the operating system under different lighting conditions, indicate safe maintenance routes, and show 3D-animated operational procedures. A Post-Study System Usability Questionnaire (PSSUQ) test should also be conducted with a number of users for statistical feedback on utilizing the BIM-based AR system to quickly locate inter-related components, such as pipes, cooling towers, and power switches. If the above-mentioned objectives are achieved and tested, maintenance efficiency and safety should be increased.

The article is organized as follows:

1. Introduction;
2. Related works: the progress of AR and its applications in FM and BIM;
3. Research purpose;
4. BIM-based AR Maintenance System (BARMS): the development process, framework, and functions of the system;
5. Experiment design: the design, task, and site of the experiment;

    5.1. Encountered problems in preliminary tests;
    5.2. Systems and location to be experimented;
    5.3. Experiment procedures;

6. Experiment result: PSSUQ analysis;
7. Discussion: further discussion of the experimental results and related issues;
8. Conclusions.

## 2. Related Works

AR has drawn broad attention in architecture, construction, and engineering (ACE) in recent years. Maintenance has become one of the frequently emphasized aspects of construction [1]. For instance, a Head-Mounted Displays (HMDs)-based AR system was developed for maintenance tasks in a complex environment [2]. A natural marker-based AR framework has also been proposed to support on-site maintenance tasks in facility management [3,4]. A novel AR-assisted system was developed to instruct and improve workflow and equipment serviceability in maintenance operations [5]. Studies have also

demonstrated that the efficiency and productivity of maintenance operations can be augmented by applying AR [6–8].

In addition to facility management, AR and BIM have been integrated with construction management through a conceptual system framework to reduce the occurrence of construction defects [9]. AR has been proposed as a way to extract information from BIM models to improve the efficiency and effectiveness of tasks performed by workers [10]. A low-cost mobile combined AR and BIM tool was developed to access information facilities [11]. Numerous studies have shown that AR can facilitate the process of mapping building documentation onto a 3D real-world entity, offering great potential for the integration of AR with BIM [12–16].

All of the above-mentioned AR research has achieved significant progress, with the majority of the maintenance applications using marker-based tracking techniques [17]. Although physical markers can be easily attached to surfaces, markers are intended to be covered by other objects according to environmental aesthetic concerns [18]. However, setting up markers can be very time-consuming in a working environment [19]. The feasibility of marker-based AR applications can be limited. Consequently, markerless AR was developed to solve this problem using SLAM technology [20,21].

Few AR systems use 3D animation functions, which have been proven to be very helpful in assembly tasks [14]. Although it is quite common to conduct maintenance tasks in relatively low light conditions, most AR applications have been tested in adequate lighting environments. The illumination of AR applications has been hardly developed for dark environments.

## 3. Research Purpose

A BIM-based markerless AR maintenance system was developed to provide a 3D animated operating instruction and a night illumination function. The application was tested for a plumbing operation system, in which the sequence of pipes, cooling towers, and power switches were included, with the aim of substantially improving the problem-solving and operational efficiency of traditional MEP maintenance.

## 4. BARMS

A building's plumbing system is usually located inside a building or in walls where a location-based AR sensor, like GPS, can hardly be applied [22,23]. For a system located outside of a building or on the roof level, a location-based AR sensor may not provide sufficient accuracy. Thus, an image-based AR was selected for these reasons to meet the need to traverse between the indoors and outdoors.

HMDs with high-quality sensors have been used for AR applications [24,25]. However, HMDs can affect the wearer's perception of their real environment, which can be hindered by safety concerns in many industrial areas. HMDs are expensive and not a daily tool to carry around. Handheld devices are more suitable for maintenance tasks [26,27]. Thus, smartphones are selected as the first priority for AR development.

ARKit was applied using the AR Software Development Kit (SDK) developed for the iPhone XS Max. BARMS comprises six modules (Figure 1). A real-time environment viewing module and an LED lighting module enable smartphone cameras to operate well in low illumination environments, with an enhanced SLAM positioning capability. The pipe-switching module selects and displays the target pipe to be serviced in AR. The 3D maintenance animation module and maintenance path module, which are closely connected to the switch module, provide safe and straightforward equipment operation guidance. The AR guide graphic module displays a concise interface for workers to retrieve maintenance-related information from an augmented information database.

The AR system possesses the following distinguishing features:

- SLAM and markerless operation in low light conditions;
- Pipe and entity information display;

- Safety maintenance route display; and
- 3D maintenance animated guide.

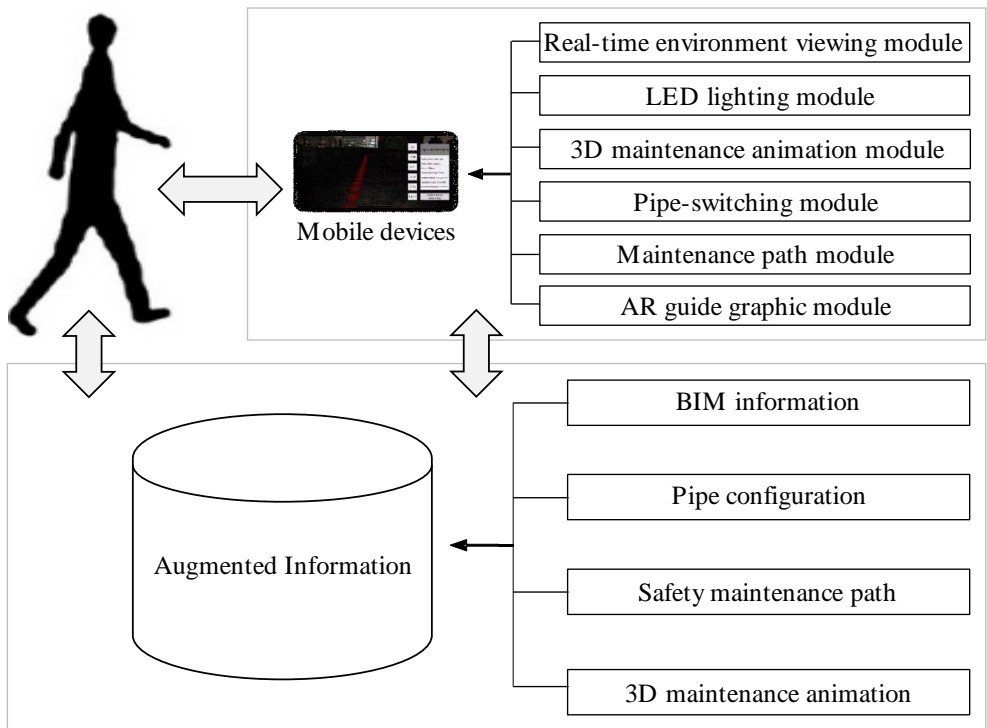

**Figure 1.** System structure and modules of BARMS.

The development process of BARMS is illustrated in Figure 2. After defining the system requirements, the 3D BIM model and related pipe information were constructed. The 3D pipe model was imported into Unity to create the AR scene. The final app was built in Xcode to code for different functions and corresponding links. An unmanned aerial vehicle (UAV), DJI Spark, was used as a measurement and drafting assistant for the 2D plan and the 3D model of the building. The BIM pipe model was constructed using Autodesk Revit. Visual Studio was utilized to code the program.

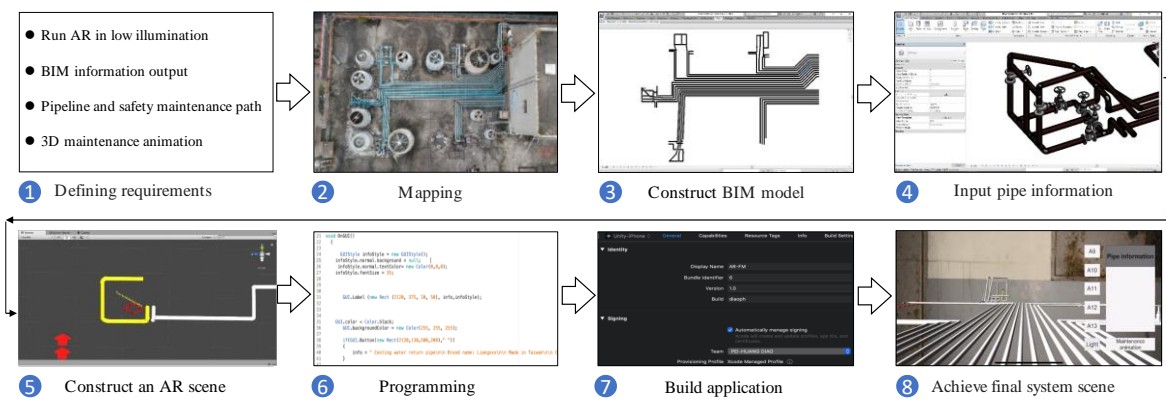

**Figure 2.** The development of BARMS.

The interactive user interface of BARMS is located on the right side of the smartphone screen. As shown in Figure 3, a user can tap on buttons in the interface to retrieve information from the augmented information database. Because the brightness of virtual objects changes based on the ambient light of an environment, safety maintenance paths will appear brighter in the daytime than at night, for a better display appearance.

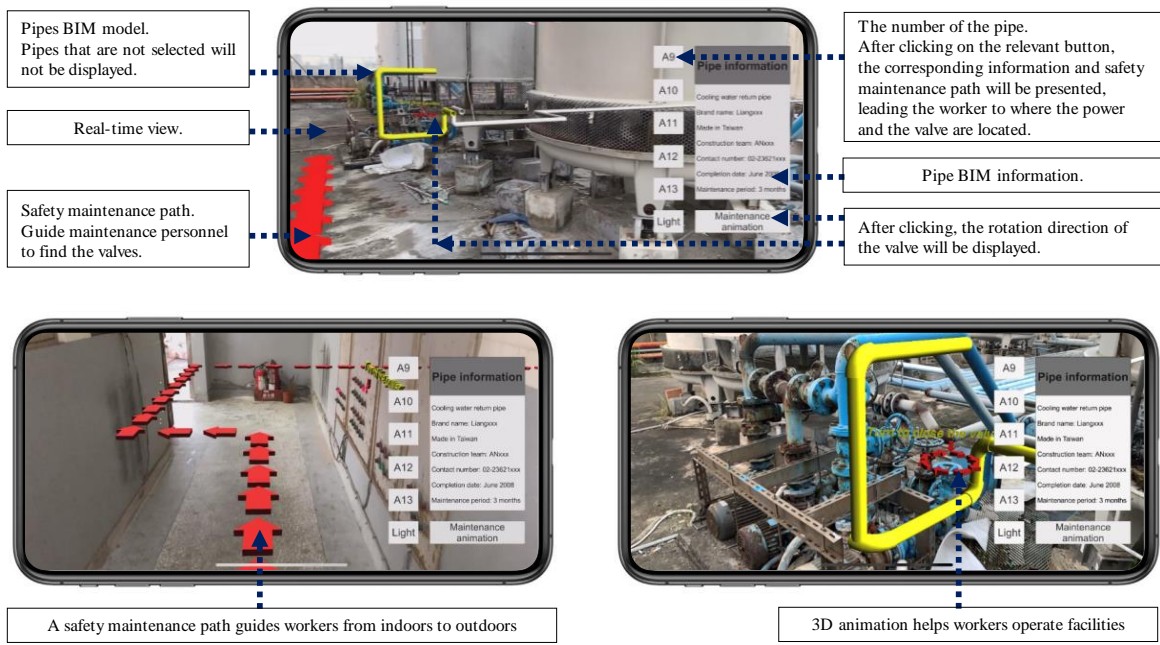

**Figure 3.** User interface of BARMS.

## 5. Experiment Design

Most campus buildings were constructed more than 29 years ago. Related MEP system installation and maintenance have been performed since then, which have resulted in a very complicated layout without a centralized office or panel that monitors or controls all on/off switches or that displays warning signals. On/off switches or power sources should either be located near target devices or be distributed by floors or zones to localize control and monitoring procedures. Conflict occurs when troubleshooting has to be carried out across several systems in which different control patterns, such as requirements, sequences, and locations, have to be simultaneously addressed at both individual and integrated levels.

This test was designed to incorporate the requirements and integration of an electrical system and a cooling system. The selected campus building, which is nine-stories high including the basement, is used by three academic departments, a university computer center, and two auditoriums. MEP systems and requirements are divided by floors or zones, where the part that was selected to be tested was the subsystem connected to the fourth floor. Due to the limited space available in the basement, the mechanical space was overcrowded with pipes, ducts, motors, condensers, etc.

University contractors were consulted several times to obtain a thorough and accurate understanding of the major MEP system settings in the building. First-hand observations were made by traveling from mechanical rooms in the basement, to duct spaces inside of the ceiling, and to shafts above the roof. Some systems were installed or updated after the original construction, using nearby limited space with a clearance that was too tight to enable a clear view of all the components. While attempting to locate the targeted room for service, the entire scope of the individual or incorporated systems, which was segmented by floors, rooms, or ceilings, was difficult to determine unless a complete set of drawings were provided with accurate updates.

### 5.1. Problems Encountered in the Preliminary Tests

A cooling tower and its related pipes, located on the roof of the campus building, were selected for the test. A BARMS system tester was supposed to apply BARMS to confirm the pipe information and to shut down the power of the cooling tower and the pump to which a valve and return pipe were connected. It was noted that the whole system could not be accessed from a control room with a centralized diagnostic panel.

During the context setup for the test, the following series of problems were encountered during preliminary tests without the assistance of BARMS:

1.  Limited scope of the incorporated systems: MEP, which was separated as an individual or an isolated system, should be considered as integrated or incorporated with the whole system in order to provide a full service function;
2.  Searching problem: the correct power switch was not able to be located;
3.  Incorrect selection: a mistake was made in selecting the same pipe on either side of a wall, from indoors to outdoors, and the correct valve of the pipe was difficult to identify among a cluster of settings;
4.  Unknown operation sequence: no background knowledge was possessed about the inter-relationship of the shut-down sequence between the valve and the cooling tower or pump;
5.  Incorrect operation sequence: the valve was shut down before the pump was powered off, which resulted in idling and overheating; and
6.  Uncertain operation details: a trial-and-error approach was attempted to shut down the pump by rotating the valve in the incorrect direction.

### 5.2. Systems and Location Tested

The following situation, system, and location specify the conditions under which BARMS was applied. The roof had a total of 10 cooling towers, 20 pipes, and a number of integrated valves (Figure 4). Pipes entered the walls of the staircase near the ground level to reach the floors below and the condensers in the basement. The cooling tower and the power panel were located outside and inside the building, respectively. In the event of an indoor leak, the maintenance worker had to determine which pipe was broken, its corresponding outdoor location, and its connected power switch on the roof level by traveling between the indoors and outdoors several times to confirm a correct match. Due to the lack of sufficient lighting devices in open air, maintenance in the evening was more difficult than that performed in the daytime.

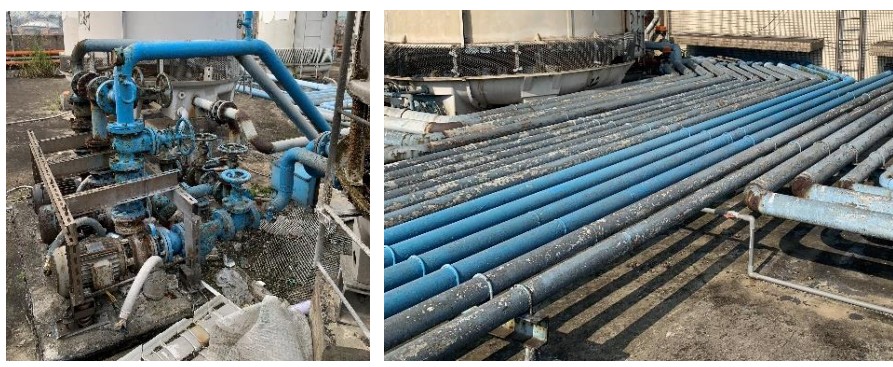

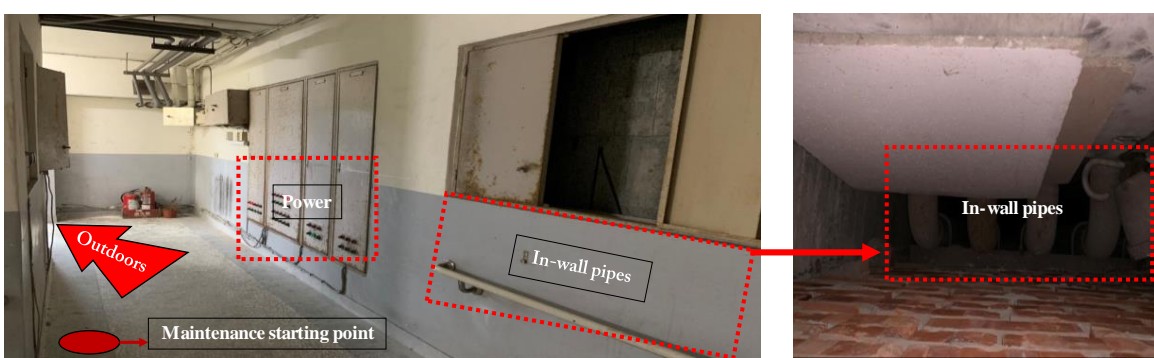

**Figure 4.** Photographs of outdoor (top) and indoor (bottom) layout to which BARMS was applied.

### 5.3. Experimental Procedures

The main test was conducted in the evening. The flowchart of operational procedures is presented in Figure 5. After an in-wall pipe was found to have severe leakage, the BARMS system tester needed to first shut down the water supply by positioning its valve either to or from the cooling tower and then shut down the corresponding power switch to the tower and controlling pump.

This test comprises six related tasks:

1. Retrieve leaking pipe-related information: the construction date and previous maintenance schedule were retrieved by taking advantage of existing BIM information;

2. Trace the corresponding location of the pipe on the other side of wall: tracking has to be performed among 20 possible pipes in order to identify the correct one. Not all of the 20 in-wall pipes are shown in the vertical shaft, i.e., some of the pipes are sealed in the walls. The tested pipe could be seen in the shaft relatively easier than the others. This constitutes one of the most difficult parts of the test, since the indoors and outdoors is separated by a wall, without any visual reference to the pipe outside. Counting the sequential layout of pipes is also impractical because some pipes are not visible in the shaft;

3. Previous research has identified a possible solution to the pipe identification problem by projecting the outline of an entity inside of a wall to the surface in order to show the correct location when observed from the outside [11]. Our study extended this concept to show the pipe on the opposite side of a space that is not directly connected. In addition, the entity-pinpointing task had to be performed for other system components, such as the power switch of an electrical system;

4. Locate the correct cooling tower to which the pipe is connected: there are two types of pipes—the cooling water supply pipe and the return pipe—which increases the complexity of accurate pipe identification;

5. Shut down the power to the cooling tower and the pump connected to the return pipe: power must be shut down before the pipe or the valve. Since the system was designed using the same switch for the tower and its related pump, both components will be powered off at the same time; and

6. Correctly shut down the valve connected to the return pipe.

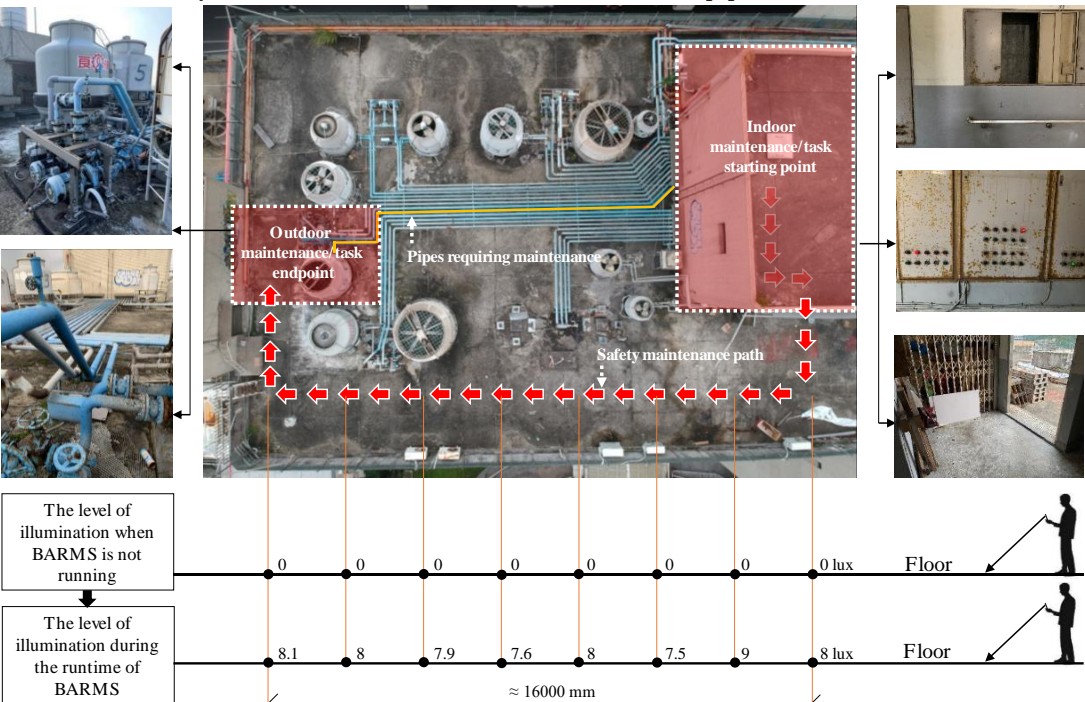

**Figure 5.** Sequence of operational procedures.

## 6. Experiment Result

A building's plumbing system is so complicated that it is often difficult for a non-professional to maintain and operate. The goal of this experiment was to assist users with different knowledge backgrounds to perform basic equipment operation and maintenance in urgent problem-solving situation using BARMS. Students of various educational backgrounds were randomly selected to participate in this experiment as testers and fill out questionnaires. In total, six doctoral students and nine master's students participated in this study, with the educational backgrounds in architecture, information engineering, and design.

The test results demonstrated that the testers who used BARMS could find the correct power switch and cooling tower that connected to the leaking pipe, and shut down the valve correctly (Figure 6). Unanimously, the testers stated that maintenance efficiency could be improved in low illumination environments by utilizing BARMS.

Screen shots of BARMS utilization are shown in Figure 7. Upon running the app, all of the regional pipes were displayed, including in-wall and downward-extending ones. The pipes were marked by 3D numbers, such as A9, A10, or A11. The tester could tap on a certain pipe number to hide unrelated pipe information. Meanwhile, the location of the power switch connected to the cooling tower and pump was indicated by a moving 3D arrow to attract the tester's attention. After turning off the power switch, the tester could walk to the corresponding cooling tower and valve by following the safe maintenance route displayed on the screen. Moreover, tapping the "Maintenance Animation" button would bring up an animated arrow around the valve showing the correct rotation direction to shut down the pump. A video sample of the test can be seen in Appendix A.

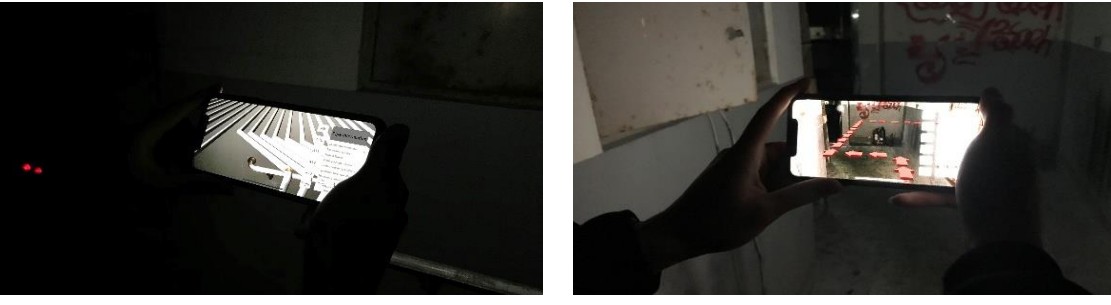

**Figure 6.** BARMS tested by a student in darkness.

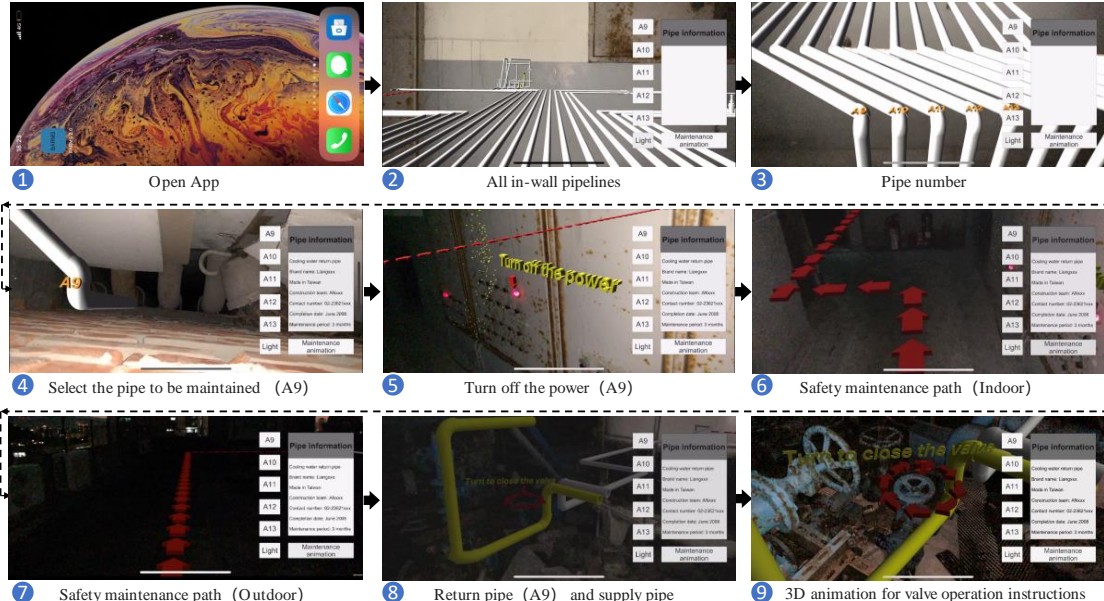

**Figure 7.** Smartphone screenshots of BARMS utilization.

The questionnaire was revised on the basis of the Post-Study System Usability Questionnaire (PSSUQ) (Appendix B). The PSSUQ has four dimensions with a total of 16 assessment items, including 1–16 items for the "overall average", 1–6 items for "system usefulness", 7–12 items for "information quality", and 13–15 items for "interface quality" (Table 1). The questionnaire items were scored on a Likert-type seven-point scale, where 1, 2, 3, 4, 5, 6, and 7 were represented by "strongly agree", "agree", "somewhat agree", "neutral", "somewhat disagree", "disagree", and "strongly disagree", respectively. Essentially, the lower the rating the higher the system's reported usability. Table 1 shows the feedback from the PSSUQ. The overall average, system usefulness, information quality, and interface quality scores were 1.74, 1.42, 1.90, and 2.00, respectively, showing high system usability [28]. Figure 8 shows more detailed scoring. Questions 7 and 8 scored relatively high because BARMS did not provide a way to correct system errors. Although there were no obvious errors in the system during the test, it was a function that the testers thought could be provided in the future. Other suggestions included adding more text to instruct the user's actions and more screen buttons for system options.

**Table 1.** The scores of the four items of the Post-Study System Usability Questionnaire (PSSUQ).

|  | Items | N | Mean | SD |
|---|---|---|---|---|
| Overall average | 1–16 | 15 | 1.74 | 0.40 |
| System usefulness | 1–6 | 15 | 1.42 | 0.30 |
| Information quality | 7–12 | 15 | 1.90 | 0.38 |
| Interface quality | 13–15 | 15 | 2.00 | 0.83 |

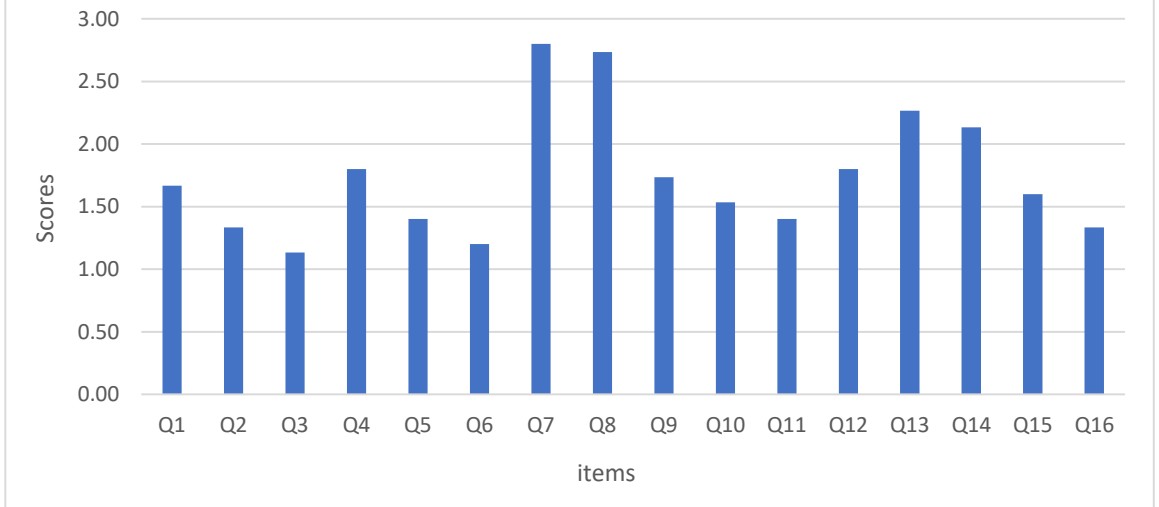

**Figure 8.** The average score for all topics in the PSSUQ.

## 7. Discussion

One of the major AR characteristics is the combination of real and virtual objects in a real environment [29,30]. The combination show a strong resemblance to that of building system maintenance, which constitutes the completion a complicated task by applying knowledge of various levels and representative forms. The diversity of mechanical equipment in building system maintenance requires hands-on experience and a thorough comprehension of the appropriate operation manuals in order to perform maintenance jobs at real sites. In addition to BIM, a task normally has to be carried out with support or resources from different fields, which possess the diverse characteristics of as-built realty, facility management, operational instruction, and working environments that could either be difficult to access or have complicated cross-space layouts throughout the entire building.

Maintenance in dark or low lumen conditions is frequently encountered on working sites where safety is paramount. Safe routes and procedures should be clearly indicated with the lighting function

clearly connected to operational signage. BARMS provides a route clear of unanticipated obstacles and an adaptable orientation for optimal nearby viewing angle. A BARMS-enabled smartphone becomes very helpful, with its easy mobility and embedded flashlight. The region of interest, lit by the smartphone flashlight, can now be connected to the entire system for a better understanding of the possibly affected components. An AR system, which can be very beneficial in working outdoors at night or in similar dark indoor situations, can now achieve an accurate level of reality that combines with the overall scope of the setting by using a commonly available illumination function. Consequently, work difficultly is significantly reduced, especially in cases involving an unfamiliar space.

BARMS provided a straightforward indication of the valve location with a clear and simple maintenance animation to lessen the workload of the testers and increase operational efficiency. The integration of virtual building component information and on-site maintenance, in reality, is critical for as-built facility management tasks. Although the BIM 3D model and its attributes contribute to complete facility information, which can be accessed by maintenance workers from cloud computing services, the real piping layout and its exact penetration points are very challenging to verify from both sides of walls. The piping layout and penetration point can now be confirmed from both sides of walls, with piping geometries facilitated and integrated with both information and reality.

Many researchers engaged in AR face the problem of poor geo-referencing [12]. Although visual tracking usually achieves the best results with low frequency motion, it is likely to fail with rapid camera movement [31]. Although BARMS can provide mediated information to assist with geometry and information mapping, drifting (0–20 cm) occurs at the far end of a pipe beginning when approximately 16.5 m away from the starting reference point. This deviation can also increase when workers approach the maintenance location from a greater distance. Currently, an auxiliary application that runs at a lower hierarchical position is used to eliminate drift at the distant location, next to the end of the pipe.

## 8. Conclusions

The system design and experimental results demonstrated that BARMS was capable of providing an illumination function in darkness, showing a safe maintenance route, projecting BIM models, confirming relative locations, and illustrating maintenance using animation. BARMS was very effective when working outdoors at night or in similar indoor low light situations by seamlessly integrating the existing conditions with the overall scope of the setting and using the readily available smartphone illumination function. The smartphone application provided a maintenance route that avoided unanticipated obstacles, and offered a viewing angle orientation with the optimal nearby. The system projected BIM models into an environment for maintenance workers and confirmed the relative locations of pipes on the other side of walls. A straightforward indication of the valve location was provided with a clear and simple maintenance animation to lessen work burden and increase operational efficiency.

The successful application of BARMS and its integration with BIM in facility management were confirmed. The PSSUQ results demonstrated that the application could identify the correct power switch and cooling tower that were connected to the leaking pipe, and assist with the proper shut down of the valve. In general, maintenance efficiency could also be improved in low illumination environments.

Future studies should address the drifting problem through better positioning accuracy by developing a well-planned network of application-anchoring locations to reduce accumulated drifting errors. It is reasonable to believe that technical solutions to these problems will be achieved by computer scientists and AR engine developers in the future.

**Author Contributions:** Software was programmed by P.-H.D. Most of the conceptualization, methodology, validation, formal analysis, investigation, resources, data curation, visualization, and writing to original draft preparation were conducted by both N.-J.S. and P.-H.D. In particular, writing to review and editing, supervision, and project administration were mainly conducted by N.-J.S.

**Funding:** This research received no external funding.

**Conflicts of Interest:** The authors declare no conflict of interest.

## Appendix A. Video Sample of the BARMS Testing

https://youtu.be/nMxNrrvvFhc.

## Appendix B. Post-Study System Usability Questionnaire (PSSUQ) for the BARMS System

1. Overall, I am satisfied with how easy it is to use this system.
2. It was simple to use this system.
3. I was able to complete the tasks and scenarios quickly using this system.
4. I felt comfortable using this system.
5. It was easy to learn to use this system.
6. I believe I could become productive quickly using this system.
7. The system gave error messages that clearly told me how to fix problems.
8. Whenever I made a mistake using the system, I could recover easily and quickly.
9. The information (such as buttons, on-screen messages, and path instructions) provided with this system was clear.
10. It was easy to find the information I needed.
11. The information was effective in helping me complete the tasks and scenarios.
12. The organization of information on the system screens was clear.
13. The interface of this system was comfortable.
14. I liked using the interface of this system.
15. This system has all the functions and capabilities I expect it to have.
16. Overall, I am satisfied with this system.

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
