# Peer review of "BIM-Based AR Maintenance System (BARMS) as an Intelligent Instruction Platform for Complex Plumbing Facilities"

_applsci, doi:10.3390/app9081592_

Round 1

Reviewer 1 Report

The submitted article covers this new research area well, although it is somewhat light in methodological approach to the study. Overall this article would be of interest to the potential audience of this publication.

Author Response

On behalf of my co-author, reviewer’s comments are highly appreciated. Corrections and explanations have been made accordingly.

Reviewer 2 Report

The paper is a honest account of a practical experiment that explains several aspects of AR development for building maintenance and FM. However, the introduction is not helpful enough: the framing of the research has to become sharper. Similarly, the discussion and conclusions need to be demarcated and different in character: the discussion considers the findings but the conclusions evaluate them. 

Author Response

(The authors gave the same response as above.)
